# Food-based multisensory stimulation ameliorates cognitive impairment after mild traumatic brain injury in male rats by modulating intestinal and brain inflammation

Yuhan Wei[1], Hanif Ullah [1], Guangneng Liao[2], Xue Xiao[3], Jiang Yao[1], Qiujing Du[1]*, Qijie Li[1]*, Ka Li[1,4]*

1 Medicine and Engineering Interdisciplinary Research Laboratory of Nursing & Materials, West China Hospital, Sichuan University/West China School of Nursing, Sichuan University, Chengdu, P.R. China, 2 Animal Experiment Center, West China Hospital, Sichuan University, Chengdu, P.R. China, 3 Laboratory of Gastric Cancer, State Key Laboratory of Biotherapy, Collaborative Innovation Center of Biotherapy and Cancer Center, West China Hospital, Sichuan University, Chengdu, P.R. China, 4 Tianfu Jincheng Laboratory, Chengdu, China

* lika127@163.com (KL); qjlbsh2019@163.com (QL); duqiujing001@163.com (QD)

## Abstract

Mild traumatic brain injury (mTBI) often leads to cognitive impairment (CI), with neuroinflammation and gut microbiota dysbiosis playing pivotal roles in its pathogenesis. This study aimed to investigate whether food-based multisensory stimulation could ameliorate cognitive deficits in mTBI rats via modulation of the gut–brain axis. Using a rat model of mTBI, we demonstrated that food-based multisensory stimulation significantly improved spatial and recognition memory, as evidenced by performance in the Morris water maze and novel object recognition tests, and reduced serum biomarkers of neurological injury (NSE, S100β). Gut microbiota analysis revealed that sensory stimuli restored microbial balance, increasing beneficial taxa such as *Ruminococcaceae* and reducing pathogenic genera such as *Alistipes*, *Prevotella*. Concurrently, senso.ry stimulation increased fecal and serum levels of short-chain fatty acids (SCFAs), particularly butyrate, which were associated with reduced gut and neuroinflammation. In vitro, butyrate supplementation exhibited significant anti-inflammatory effects, promoting M2 microglial polarization and reducing pro-inflammatory cytokines (TNF-α, IL-1β). Histological analyses further revealed neuroprotective effects, preserving neuronal density in the hippocampus and cortex. These findings suggest that multisensory stimulation may mitigate CI post-mTBI by restoring gut microbiota homeostasis, enhancing butyrate production, and attenuating neuroinflammation. This non-invasive approach holds promise for cognitive rehabilitation in patients with mTBI, although further research is needed to elucidate its long-term effects and translational potential.

**Data availability statement:** All relevant data are within the paper and its Supporting Information files.

**Funding:** This research was funded by the Regional joint key projects of NSFC (No. U22A20334), and Sichuan Province Science and Technology Support Program (2024NSFSC0592).

**Competing interests:** NO authors have competing interests.

## Introduction

Mild traumatic brain injury (mTBI) is one of the most common neurological injuries in clinical practice, typically caused by external forces acting on the head [1]. Although the acute symptoms of mTBI are relatively mild, its potential long-term effects, particularly cognitive impairment (CI), can have profound impacts on patients' quality of life [2]. The mechanisms underlying CI following mTBI involve multiple factors, including neuroinflammation, neuronal damage, and oxidative stress, etc., with neuroinflammation being the primary factor [3]. In recent years, there has been increasing evidence that the gut-brain axis plays a key role in traumatic brain injury (TBI)-related outcomes. Experimental models of repeated mild traumatic brain injury (rmTBI) have shown that gut-targeted interventions, such as quercetin or chitosan lactate, can attenuate rmTBI-induced anxiety- and depression-like behaviors and cognitive dysfunction, as well as restore the expression of tight junction proteins and synaptic markers and modulating levels of inflammatory cytokines and short-chain fatty acids [4,5]. Taken together, these findings suggest that dysregulation of the gut microbiota and its metabolites after traumatic brain injury may exacerbate both neuroinflammation and chronic systemic inflammation, and that modulation of the gut-brain axis may represent a promising therapeutic strategy. However, the mechanisms by which gut microbiota dysbiosis after mTBI influences neuroinflammation and cognitive function are not yet fully understood, particularly how interventions targeting the "gut–brain axis" can alleviate neuroinflammation and subsequently improve CI remains an urgent issue to be addressed [6].

Multisensory stimulation has broad potential applications in the field of neurological rehabilitation in recent years as a non-invasive neuromodulation technique [7]. In contrast to oral nutraceuticals or probiotics, multisensory approaches act primarily through sensory and neural pathways but may secondarily influence the microbiota–gut–brain axis. Sensory stimulation not only directly affects the central nervous system, promoting neuroplasticity and functional recovery, but may also indirectly influence the function of the "gut–brain axis" by modulating the gut microbiota and its metabolites [8]. Research indicates that sensory stimulation can affect the composition and metabolic activities of the gut microbiota by regulating the autonomic nervous system and the hypothalamic-pituitary-adrenal (HPA) axis, thereby improving gut barrier function and inflammatory status [9,10]. Additionally, SCFAs such as butyrate, produced by gut microbiota metabolism, can cross the blood-brain barrier, exerting anti-inflammatory and neuroprotective effects, thus reducing brain inflammation and improving cognitive function [11]. However, the mechanism by which multisensory stimulation influences neuroinflammation and subsequently improves CI after mTBI through the modulation of gut microbiota and its metabolites has not been systematically studied.

As a distinct form of multimodal input, sensory stimulation from food integrates visual, olfactory, gustatory, and oral tactile features; it is closely linked to motivational and reward systems and may therefore exert unique effects on cognitive enhancement. Given that food intake naturally engages gut motility, secretion, and microbial metabolism, food-related sensory cues may be particularly well-suited to influence

both central cognitive circuits and the microbiota–gut–brain axis. Clinical studies indicate that food recognition is relatively preserved in Alzheimer's disease and primary progressive aphasia, suggesting that food-related cues may be more resistant to degradation [12]. Another study reported that, among patients with mild CI or mild dementia, a multisensory food-based art therapy—entailing the viewing and tasting of foods with varied colors and textures—can positively influence cognitive, emotional, and social functioning [13]. There is also evidence that food cues elicit attentional biases, enhance hedonic appraisal and ingestive motivation, and are accompanied by widespread changes in neural activity, suggesting that food-related sensory stimuli are coupled to cognitive processes such as attention and reward evaluation [14]. However, the extent to which these effects generalize to broader cognitive functions in non-food contexts remains to be determined. In addition, no studies have evaluated food-based multisensory stimulation to ameliorate cognitive deficits after traumatic brain injury, and its efficacy and applicability remain unclear.

Based on this, the present study aimed to investigate the effects of food-based multisensory stimulation on cognitive function in rats with mTBI, and to elucidate, from the perspectives of gut microecology and neuroinflammation, the molecular mechanisms by which it improves post-mTBI cognitive function via modulation of the gut–brain axis. We hypothesized that food-based multisensory stimulation could modulate the composition of the gut microbiota, promote the proliferation of beneficial microbiota and the production of SCFAs (particularly butyrate), thereby improving gut barrier function, reducing intestinal and cerebral inflammatory responses, and ultimately alleviating cognitive dysfunction following mTBI.

## Material and methods

### Ethical statement

This study was sanctioned by the Experimental Animal Ethics Committee of West China Hospital, Sichuan University (Approval No. 20250314034). All experimental procedures were carried out in accordance with the ARRIVE guidelines, and all possible efforts were made to minimize animal suffering [15]. On the last day of the experiment, all animals were euthanized by cervical dislocation under general anesthesia.

### Mild traumatic brain injury model in rats

Male Sprague-Dawley (SD) rats weighing 200–300 grams and aged 6–8 weeks were sourced from Chengdu Enswell Biotechnology Co., Ltd. (Chengdu, China). The housing environment was maintained under strict control for temperature and humidity, and all subjects underwent a one-week acclimatization period before surgical procedures, during which they were exposed to a 12-hour light/dark cycle and had unrestricted access to food and water. Upon receipt, 24 animals were randomly allocated into three experimental groups (n = 8 per group), with cohabitation within cages restricted to members of the same group. Group assignments were performed by an investigator who was not involved in subsequent behavioral testing or data analysis, and the investigator who conducted the behavioral testing and data analysis was blinded to the group assignments.

The mTBI induction protocol was adapted from Feeney's method [16]. All surgical interventions were performed under sterile conditions while employing temperature-controlled blankets to maintain the body temperature of the rats at 37−39°C. Anesthesia was initiated using isoflurane at a concentration of 3.5% and sustained at levels of 1.5%−2%. Preoperative skin preparation involved sequential cleansing with Betadine Scrub (7.5%), 70% alcohol, and Betadine Solution (5.0%). A midline incision was made along the scalp, followed by the attachment of a stainless-steel disc (dimensions: 10 mm × 2 mm) using dental acrylic. Following positioning on a foam pad, an impactor weighing 450 grams and having a diameter of 10 mm was dropped from a height of 1.0 m to simulate mTBI [17,18]. Postoperatively, the stainless-steel disc was removed, and the wound was closed with sterile sutures. In the sham-operated group, the procedure, including the duration and conditions of isoflurane anesthesia, mirrored that of the mTBI groups up until the infliction of the weight drop. After surgery, animals were placed in a 37°C incubator in a supine position. One group subjected to mTBI received audio-visual and olfactory stimulation, whereas the remaining groups did not receive any intervention.

Interventions commenced immediately post-injury and continued for seven days, operating around the clock. This 7-day experimental protocol was designed to cover the acute and early subacute phases following mTBI. During this period, gut barrier dysfunction and alterations in the gut microbiota are most pronounced [19], which is consistent with the 7-day treatment regimen employed in previous gut-targeted interventions in rmTBI models [4]. The detailed experimental timeline and sequence of all procedures are illustrated in the study design diagram shown in Fig 1.

## Based on the multisensory stimulation afforded by food

The food-based multisensory stimulation device used in this animal study was derived from the integrated audio-visual-olfactory virtual reality false feeding device previously researched, developed, and designed by our team. This device simulates the oral feeding state by integrating audio-visual-olfactory sensory stimuli to stimulate brain neurons, thereby inducing related physiological responses and functional expression and compensating for the shortcomings of tube feeding or simple oral enteral nutrition supply methods [20].

In this study, the device for food-based multisensory stimulation consisted of a transparent, sealed acrylic box (9.6 × 6.3 × 5.5 cm). Based on pilot tests of food preference, pet biscuits and chicken jerky that elicited the strongest approach and sniffing behavior were selected. According to the manufacturer, the biscuits contained (%, w/w): crude protein ≥ 8.0, crude fat ≥ 1.0, crude ash ≤ 4.0, crude fiber ≤ 1.0, moisture ≤ 15.0; the chicken jerky contained: crude protein ≥ 70.0, crude fat ≥ 3.0, crude ash ≤ 10.0, crude fiber ≤ 10.0, moisture ≤ 10.0. These items were placed inside the box together with cotton balls bearing a sweet food aroma. The odor solution was diluted 1:50 (v/v; 1 part stock in 49 parts distilled water), and 3–5 drops of the diluted solution were applied to the cotton balls before each session. Small holes (≈2–3 mm diameter) were made in the box walls to allow odor emission. For auditory stimulation, recordings of rats chewing food were played from a recorder placed next to the box at approximately 50–60 dB at the level of the animal. Rats were exposed only to the sensory properties of food; they had no physical access to the food items at any time during the experiment. Interventions commenced immediately post-injury and continued for seven days, operating around the clock. A schematic illustration of the multisensory stimulation setup is provided in Fig 1.

## Morris water maze (MWM)

Minor refinements were implemented to enhance the methodologies previously employed in the Morris Water Maze (MWM) experiments [21]. Before undergoing surgery, all rats underwent a preliminary training regimen consisting of ten sessions over two days. During these sessions, the rats were randomly positioned in the water and tasked with locating a submerged platform. Only those rats that successfully identified the platform at least nine times out of ten attempts, with an average escape latency not exceeding 120 seconds, were selected for subsequent experimental procedures.

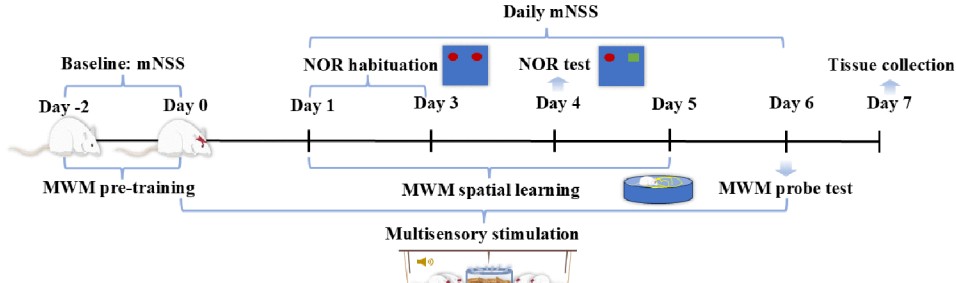

**Fig 1. Experimental timeline and schematic of the food-based multisensory stimulation apparatus.**

Post-surgical MWM assessments commenced on the first day following the modeling procedure, taking advantage of the animals' heightened alertness and motivation during the morning hours. From post-modeling days 1–5, spatial navigation tests were conducted, comprising four trials per day with 20-minute intervals between each session. These trials documented the rats' swimming paths and the initial times required to locate the platform. On day 6, a spatial memory retention test was administered; specifically, 24 hours after the final training session, the platform was removed from the maze, and the rats were placed in the third quadrant. Over a period of 90 seconds, the rats' swim trajectories, velocities, and time spent in the target quadrant were recorded using specialized computer software. Additionally, the frequency of crossings over the former platform location was documented to assess spatial memory retention. To control for potential motor confounds, swimming speed was recorded for each trial and compared across groups. This rigorous protocol ensured precise evaluation of cognitive function alterations following the experimental interventions.

### Novel object recognition

The Novel Object Recognition (NOR) experiment was conducted over four days, starting the day after modeling, with all sessions performed in the morning to align with peak rat exploratory activity. The protocol included three phases: adaptation, familiarization, and testing. During the three-day adaptation phase, rats were placed in an empty test box (50 × 36 cm) for 10 minutes each day. On the fourth day, the familiarization phase involved placing two identical objects in the box and recording each rat's exploration time for 5 minutes. After a 30-minute interval in feeding cages, rats were returned to the test box for the testing phase, in which one familiar object was replaced with a novel object differing in shape and color. Exploration times for both objects were recorded. Cognitive performance was assessed using the Relative Discrimination Index (DI), calculated as DI = (N − F)/ (N + F) * 100%, where N is the exploration time for the new object and F is for the familiar object [22]. To avoid location bias, the positions of the familiar and new objects were alternated across trials. A DI exceeding 50% indicates normal recognition memory [23], providing a clear metric for evaluating cognitive function. In addition, total exploration time was analyzed to account for possible differences in locomotor activity or motivation across groups. This streamlined protocol ensures precise evaluation of learning and memory capabilities.

### Modified neurological severity scores (mNSS)

The Modified Neurological Severity Score (mNSS) was employed in accordance with established protocols [24], to evaluate neurological function. This assessment encompasses sensory function, motor performance, balance, and reflexes. Neurological deficits were quantified on a scale from 0 to 18, where a higher score indicates a more severe impairment. The mNSS scores were recorded before the induction of mTBI and subsequently on days 1–6 post-injuries. This systematic scoring facilitates the precise monitoring of neurological changes over time.

### Enzyme-Linked Immunosorbent Assay (ELISA)

Venous blood samples were collected from rats prior to injury and on post-injury days 2, 4, and 6, with repeated sampling from the same animals at each time point; blood samples from all groups were collected at identical time points. Serum was separated and analyzed by enzyme-linked immunosorbent assay (ELISA). Following the manufacturer's instructions, samples were diluted as required and assayed using commercial ELISA kits to determine concentrations of neuron-specific enolase (NSE), S-100 β protein, and D-lactate (D-Lac). The ELISA kits were obtained from Byabscience (Nanjing, China), including those specific for rat NSE, S-100 β, and D-Lac.

### Quantitative real-time polymerase chain reaction (q-PCR)

On the seventh day post-modeling, brain and intestinal tissues were collected from the rats. mRNA levels of TNF-α, IL-6, and IL-1β were analyzed in the jejunum and ileum, while TNF-α, IL-6, IL-1β, IL-4, and IL-10 were assessed in the peri-lesional cerebral cortex (CTX) and microglial cells using qPCR. Total RNA was extracted using TRIzol Reagent (CWBIO)

and reverse-transcribed with the Quantscript RT Kit (Tiangen Biotech Co.). PCR reactions were performed in a 20 µL volume using the SYBR Premix Ex Taq Kit (Applied Biological Materials Inc.) on a 7900 HT Fast Real-Time PCR System (Applied Biosystems). The cycling conditions were: initial denaturation at 95 °C for 3 minutes, followed by 40 cycles of 95 °C for 15 seconds and 60 °C for 30 seconds. Primers (**Table 1**) were designed with Primer Express software and synthesized by Sangon Biotech.

## HE and Nissl staining

Hematoxylin and Eosin (H&E) staining was performed to assess the morphology of intestinal tissues. Jejunum and ileum samples were fixed in 4% paraformaldehyde, embedded in paraffin, and sectioned at a thickness of 5 µm. Sections were stained with hematoxylin and eosin and examined using a light microscope. For Nissl staining, brain and hippocampal tissues were fixed in 4% paraformaldehyde for 24 h and cryoprotected in 30% sucrose for 48 h. Frozen sections (4 µm) were cut at −20 °C with a Leica CM1950 cryostat, air-dried, and stained with 1% toluidine blue for 5 min. Sections were then rinsed with distilled water, dehydrated through graded ethanol, cleared in xylene, and mounted for observation under a Leica DM2500 microscope. Quantification of positively stained areas was performed using ImageJ software [25].

## Immunofluorescence

Immunofluorescence staining was carried out to detect M1 (F4/80⁺CD86⁺) and M2 (F4/80⁺CD206⁺) microglia in the peri-lesional cerebral cortex (CTX). Sections were incubated with primary antibodies against F4/80 (1:400, 28463-1-AP), CD86 (1:200, ER1906-01), and CD206 (1:200, ab400631). Nuclei were counterstained with 4′,6-diamidino-2-phenylindole (DAPI, G1012). Fluorescent images were acquired and analyzed using a TCS-SP8 STED 3X confocal microscope.

## 16SrRNA sequencing and analysis

Fresh fecal samples were collected from rats on post-injury day 6 for 16S rRNA sequencing. Genomic DNA was isolated using the DNeasy PowerSoil Kit (Qiagen, Hilden, Germany). DNA concentration and purity were assessed with a Nano-Drop 2000 spectrophotometer (Thermo Fisher Scientific, Waltham, MA, USA) and verified by agarose gel electrophoresis. The V3–V4 region of the 16S rRNA gene was amplified by PCR using barcoded universal primers 343F and 798R with Tks Gflex DNA Polymerase (Takara) [26]. PCR products were sequenced on the Illumina iSeq 100 platform [27]. Raw FASTQ files were first processed with Cutadapt to remove adapter sequences. Paired-end reads were then trimmed, quality-filtered, denoised, merged, and screened for chimeras using the DADA2 pipeline implemented in QIIME 2 with default parameters. This workflow generated representative sequences and an amplicon sequence variant (ASV) abundance table. Microbial diversity was evaluated by calculating alpha-diversity indices, and beta-diversity was assessed using unweighted UniFrac distances with principal coordinate analysis (PCoA). Phylogenetic trees were constructed within the QIIME 2 environment. Differential analysis was performed using ANOVA, Kruskal-Wallis, t-tests, and Wilcoxon tests through the R package.

**Table 1. Sequences of primers for real-time quantitative PCR.**

| Gene | Species | Forward primer (5' to 3') | Reverse primer (5' to 3') |
|---|---|---|---|
| TNF-α | rats | GATCGGTCCCAACAAGGAG | CTTGTCACTCGAGTTTTGAGA |
| IL-6 | rats | GATACCACCCACAACAGACCA | CAGAATTGCCATTGCACAACTC |
| IL-1β | rats | GCTTTCGACAGTGAGGAGAAT | GCTGCTGTGAGATTTGAAGCT |
| IL-4 | rats | CCACGGAGAACGAGCTCATC | ACCGAGAACCCCAGACTTGTT |
| IL-10 | rats | GCTGCCTTCAGTCAAGTGAAG | CTGACAAGGCTTGGCAACC |
| ACTB(β-actin) | rats | GACGGTCAGGTCATCACTATC | CAACGTCACACTTCATGATGGA |

### Extraction and measurement of short-chain fatty acids (SCFAs) levels

On post-modeling day six, SCFAs in rat fecal and serum samples were measured. Fecal samples (0.2 g) were homogenized in 1 mL ultrapure water containing 2,2-dimethylbutyric acid as an internal standard, then centrifuged at 12,000 rpm for 10 minutes at 4 °C. The supernatant was mixed with 10 mL of 50% sulfuric acid, 2 mL diethyl ether, and 0.5 g sodium sulfate, vortexed for 1 minute, and centrifuged at 5,000 rpm for 10 minutes at room temperature. The ether layer was analyzed by GC-MS (5977B GC-MSD, Agilent Technologies) using an HP-free fatty acid column with helium as the carrier gas, increasing the oven temperature from 90 °C to 180 °C at 15 °C/min. Serum SCFAs were extracted using ethanol and 5% trichloroacetic acid, following a similar procedure. Data were analyzed using MassHunter Workstation software.

### Cell culture and treatment

Microglial cells (BV2 murine microglial cell line) were maintained at 37 °C in a humidified incubator with 5% $CO_2$ in Dulbecco's Modified Eagle's Medium (DMEM; Gibco, Franklin Lakes, NJ, USA) supplemented with 10% fetal bovine serum and antibiotics (100 U/mL penicillin and 100 μg/mL streptomycin; Gibco). For experimental treatments, some cells were exposed to lipopolysaccharides (LPS) at a concentration of 200 μM (Sigma-Aldrich), while others were treated with butyrate at 5 mM simultaneously [28]. A control group of cells remained untreated. Drug administration occurred when cell confluence reached approximately 80%. Following a 24-hour incubation period, the cells were harvested for qPCR analysis.

### Statistical analysis

Data are expressed as mean ± standard deviation (SD). Differences between groups were analyzed by one-way ANOVA followed by Tukey's post hoc test for multiple comparisons within each predefined family of related measures. Pearson correlation analysis was used to assess relationships between variables. A p-value < 0.05 was considered statistically significant. Analyses were performed using Origin 2021 (OriginLab, USA) and SPSS 25.0 (IBM Corporation, Armonk, NY, USA). No animals, experimental units, or data points were excluded from the analyses.

## Results

### Food-based multisensory stimulation may prevent CI and neurological deficits in mTBI rats

In this study, we systematically evaluated the impact of food-based multisensory stimulation on cognitive and neurological outcomes in a model of mTBI using rats. Our findings indicated that mTBI-induced impairments were reflected in the MWM performance, where affected animals exhibited less efficient navigation trajectories compared to sham controls. These deficits were partially mitigated by exposure to the aforementioned sensory stimuli (Fig 2 A, B). Over the course of training, there was a general reduction in escape latency across all experimental groups; however, starting from the fourth day, sensory stimulation significantly accelerated this decrement in mTBI rats, with effects persisting through the fifth day of assessment (F = 359.04; P < 0.05) (Fig 2 C). Furthermore, mTBI rats exhibited a reduced number of platform crossings and decreased time spent in the target quadrant compared with sham-operated controls, indicating impaired spatial memory retention. Sensory stimuli intervention significantly increased the number of platform crossings (F = 13.35, P < 0.05) (Fig 2 D) and prolonged the time spent in the target quadrant (F = 152.107, P < 0.05) (Fig 2 E), whereas swimming velocity was not affected (F = 0.786, P > 0.05) (Fig 2 F). In the context of recognition memory, as measured by the discrimination index (DI) in a novel object recognition task, mTBI led to a significant decrease in DI values below 50%, indicative of impaired recognition memory. This deficit was reversed following sensory stimuli treatment (F = 171.1, P < 0.05) (Fig 2 G), although no notable differences were observed in the overall exploration duration among groups (F = 0.2597, P > 0.05) (Fig 2 H). Additionally, neurological function was assessed post-mTBI via measurement of mNSS scores, serum NSE levels, and S-100β protein concentrations. Following injury, an initial increase in these biomarkers was followed by a gradual decline, suggesting a trend towards neurological recovery over time. Treatment with food-based multisensory

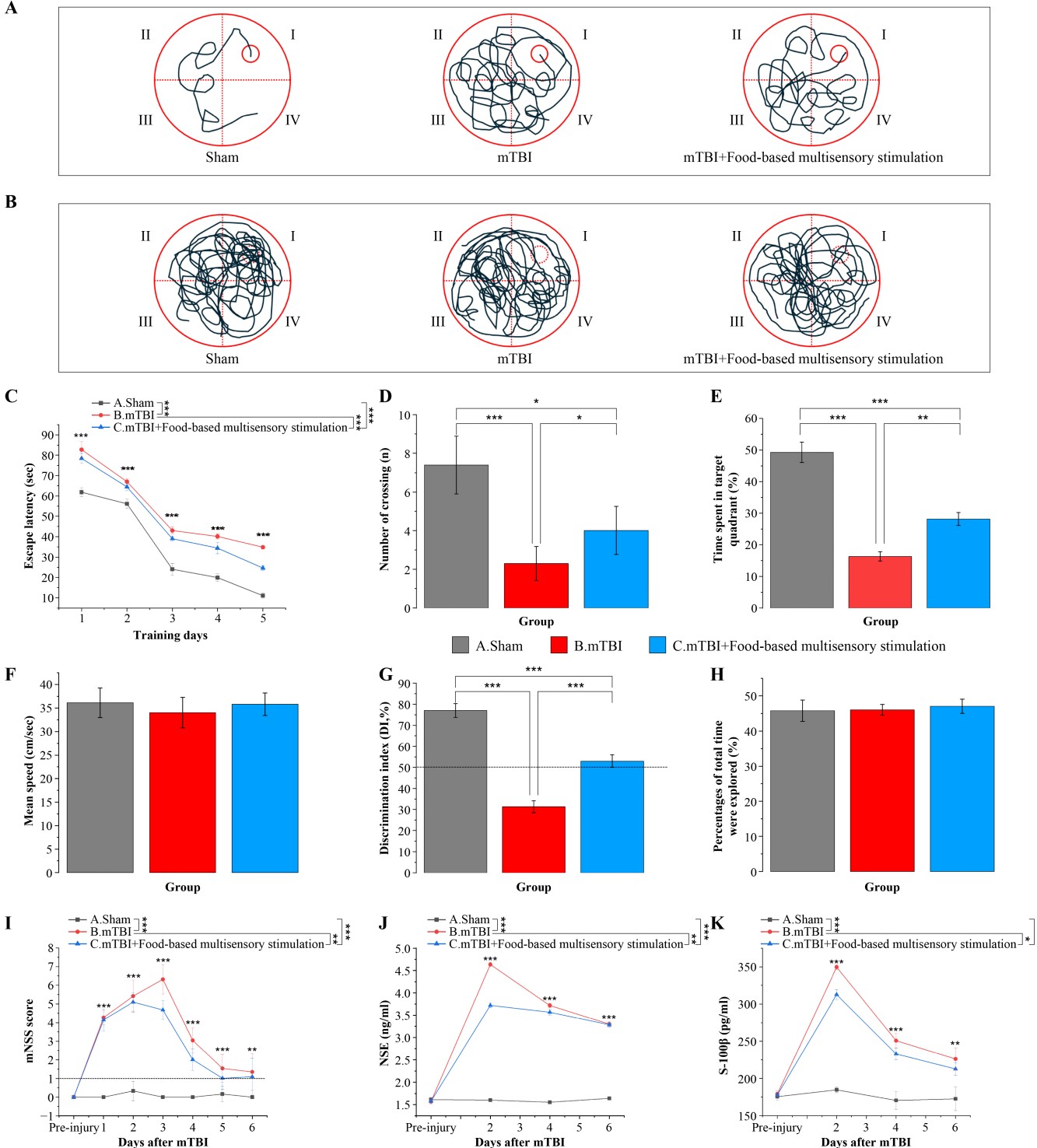

**Fig 2. Food-based multisensory stimulation may prevent cognitive impairment and neurological deficits in mTBI rats. A.** Trajectory of positioning navigation test during the water maze test. **B.** Trajectory of spatial probe test during the water maze test. **C.** Escape latency during the water maze test. **D.** The number of crossings of the target platform during the water maze test. **E.** Time spent in the target quadrant during the water maze test. **F.** Mean swimming speed during the water maze test. **G.** Discrimination index during the novel object recognition test. **H.** Percentages of total time were

explored during the novel object recognition test. **I.** Trend of modified Neurological Severity Score (mNSS) after mTBI in each group. **J.** Trend of serum Neuron-specific enolase (NSE) concentration after mTBI in each group. **K.** Trend of serum S-100 β protein concentration after mTBI in each group. N = 8/group, one-way ANOVA with Tukey post hoc test. Data are presented as mean±SD. *$p < 0.05$, **$p < 0.01$ or ***$p < 0.001$ versus mTBI group.

stimulation was associated with significantly decreased mNSS scores (F = 127.36, P < 0.05) (Fig 2 I) and reduced serum NSE (F = 2159.6, P < 0.05) (Fig 2 J) and S-100β levels (F = 1024.6, P < 0.05) (Fig 2 K) in mTBI rats. Collectively, our data demonstrated that sensory stimulation was associated with attenuated cognitive impairments and neurological dysfunction in mTBI rats.

### Food-based multisensory stimulation may modulate the intestinal microbiota in rats with mTBI

To further investigate the influence of food-based multisensory stimulation on the gut microbiota composition in rats subjected to mTBI, we conducted a genus-level analysis of the fecal microbial community six days post-injury. Our results indicated that although there were no significant differences in alpha diversity indices among the groups (P > 0.05) (Fig 3A), distinct alterations in the gut microbiota structure were observed. Specifically, compared to the sham control group, the mTBI groups exhibited a shift characterized by a reduction in beneficial bacterial taxa and an increase in potentially pathogenic taxa. This was evidenced by a decreased relative abundance of *Ruminococcaceae* and an increased relative

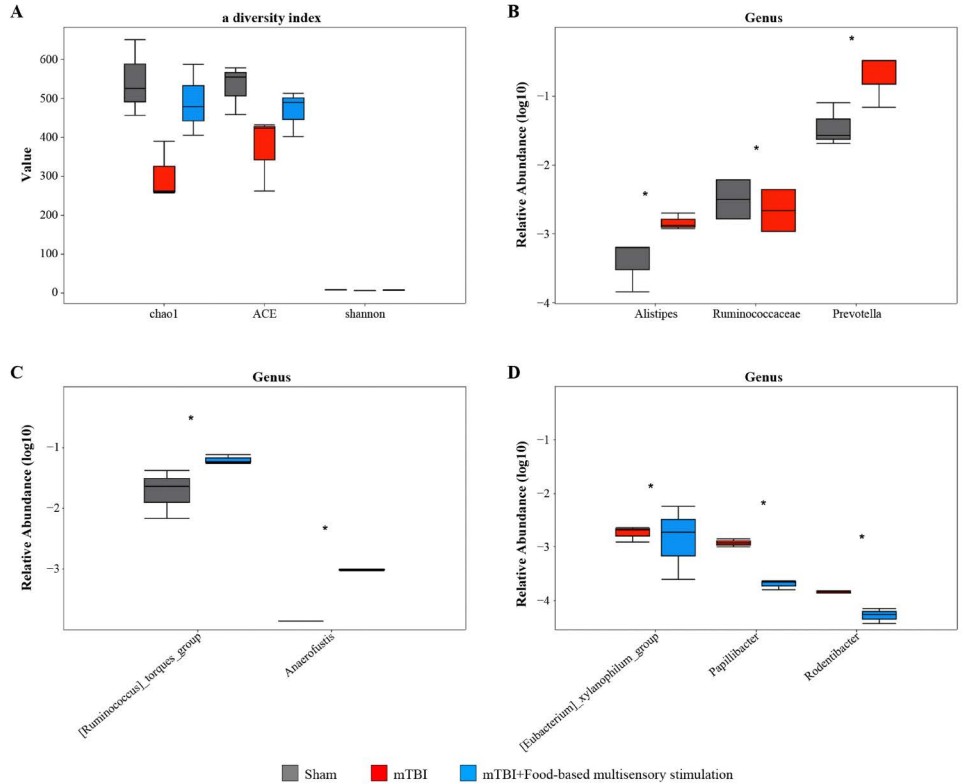

**Fig 3. Food-based multisensory stimulation may modulate the intestinal microbiota in rats with mTBI. A.** Differences in α diversity among each group, analyzed by 16S rRNA sequencing (Kruskal-Wallis H with Dunn's post hoc test). **B-D.** Analysis of differentially abundant gut microbiota at the genus level among each group, analyzed by 16S rRNA sequencing (the unpaired Wilcoxon rank-sum test). N = 8/group, data are presented as mean±SD or median (IQR). *$p < 0.05$, **$p < 0.01$ or ***$p < 0.001$.

abundance of genera such as *Alistipes*, *Prevotella*, *[Ruminococcus]_torques_group*, and *Anaerofustis* (P<0.05) (Fig 3B, C). Interestingly, the administration of food-based multisensory stimulation partially reversed mTBI-induced dysbiosis. This was reflected in a reduction in the relative abundances of several potentially harmful bacteria, including *[Eubacterium_ xylanophilum_group]*, *Papillibacter*, and *Rodentibacter* (P<0.05) (Fig 3 D). These findings suggested that sensory stimulation may play a role in restoring the balance of the gut microbiota in mTBI-affected rats, potentially through mechanisms involving the enhancement of beneficial bacterial populations and the suppression of detrimental ones.

## Food-based multisensory stimulation could reduce intestinal inflammation and damage in mTBI rats

We investigated the effects of food-based multisensory stimulation on intestinal inflammation in rat models of mTBI. Compared with sham controls, mTBI significantly increased TNF-α (F=6.12), IL-6 (F=52.37), and IL-1β (F=42.15) mRNA levels in the jejunum (P<0.05) (Fig 4 C–E), as well as TNF-α (F=9.74), IL-6 (F=6.63), and IL-1β (F=12.35) mRNA levels in the ileum (P<0.05) (Fig 4 F–H). Notably, exposure to food-based multisensory stimulation was associated with a reduction in ileal inflammation, indicated by decreased mRNA levels of TNF-α and IL-1β (F=12.35, P<0.05; F=9.74, P<0.05) (Fig 4 F, H), whereas no significant improvement was observed in the jejunum (IL-1β: F=42.15; IL-6: 52.37; TNF-α: F=6.12; all P>0.05) (Fig 4 C–E). Furthermore, we evaluated the impact of these stimuli on the integrity of the intestinal barrier. Histopathological examination revealed that mTBI resulted in more severe disruption of the villous structure in both the jejunum and ileum, characterized by villous fragmentation, epithelial cell shedding, and dilatation of the central lacteal compared to sham controls (Fig 4 A). Treatment with food-based multisensory stimulation was found to ameliorate the structural integrity of the villi in both segments of the intestine (Fig 4 A). To further assess intestinal barrier function, serum D-Lac levels, a sensitive marker of intestinal permeability, were measured. Post-mTBI, there was a progressive increase in serum D-Lac levels over time, indicative of increasing intestinal permeability. However, rats subjected to sensory stimuli exhibited reduced serum D-Lac levels by day 6 post-injury (F=10.92, P<0.05) (Fig 4 B), suggesting an improvement in intestinal barrier function. These findings collectively indicated that food-based multisensory stimulation may have potential benefits that are associated with reduced intestinal inflammation and barrier dysfunction following mTBI.

## Food-based multisensory stimulation may reduce brain inflammation in mTBI rats through the metabolite butyrate

Subsequently, we examined the effects of food-based multisensory stimulation on brain inflammation in rat models of mTBI. Our results demonstrated that these stimuli promoted a shift from M1 to M2 microglial polarization, characterized by a reduction in M1 microglia and an increase in M2 microglia (F=9.715, P<0.05) (Fig 5 A-C). Additionally, the mRNA levels of pro-inflammatory cytokines TNF-α (F=24.87, P<0.05) and IL-1β (F=23.15, P<0.05) were decreased, while those of anti-inflammatory cytokines IL-4 (F=328.45, P<0.05) and IL-10 (F=184.27, P<0.05) were increased in the peri-lesional CTX following sensory stimulation (P<0.05) (Fig 5 D-H). To assess the impact of these stimuli on neuronal health and survival, Nissl staining was performed. The analysis revealed significant neuronal damage in the peri-lesional CTX (F=23.18, P<0.05) and hippocampal CA3 regions post-mTBI (F=27.43, P<0.05), whereas sensory stimuli provided neuroprotective effects in these areas, though no significant impact was observed in the CA1 region (Fig 5 IL). Pearson correlation analysis indicated a positive correlation between neuronal density in the peri-lesional CTX and hippocampal CA3 regions with cognitive behavioral indices such as DI, number of platform crossings, mean speed, and time spent in the target quadrant, while a negative correlation was observed with escape latency (Fig 5 M). No significant correlation was found with total exploration time. These findings suggested that neuronal density in the peri-lesional CTX and CA3 regions is closely associated with cognitive performance in mTBI rats. Reduced neuronal density in these regions may contribute to cognitive decline, while sensory stimuli may partially mitigate these effects. To further elucidate the mechanisms underlying the beneficial effects of sensory stimuli on brain inflammation and cognitive dysfunction, targeted metabolomic analyses of fecal and serum samples were conducted six days post-injury. We observed that sensory stimuli increased the levels of acetate

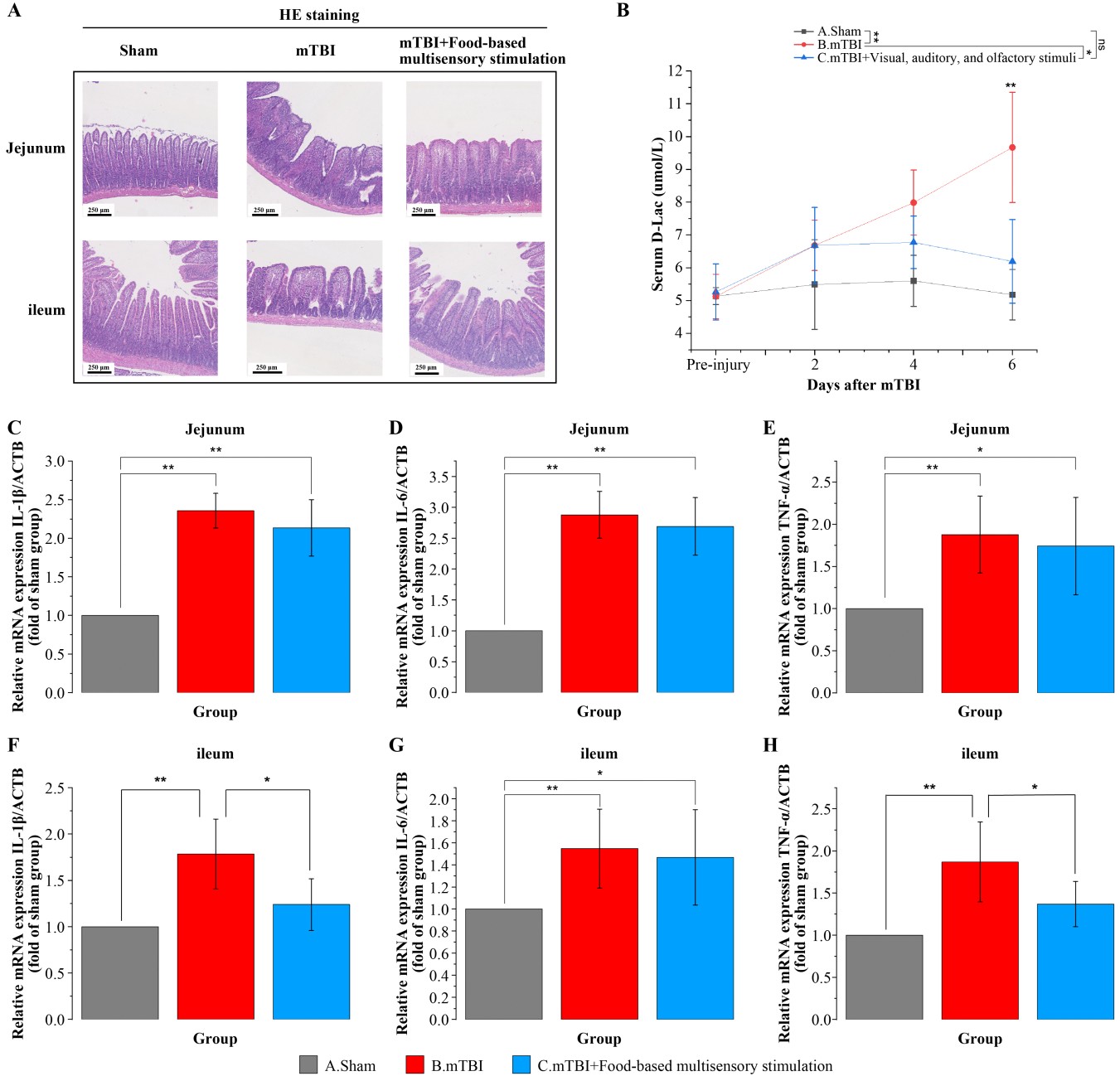

**Fig 4. Food-based multisensory stimulation could reduce intestinal inflammation and damage in mTBI rats. A.** Representative images of HE staining in the jejunum and ileum in each group. **B.** Trend of serum D-Lac concentration after mTBI in each group. **C-H.** Quantitative PCR detection of TNF-α, IL-6, and IL-1β in the jejunum and ileum in each group. N = 8/group, one-way ANOVA with Tukey post hoc test. data are presented as mean±SD. *$p < 0.05$, **$p < 0.01$ or ***$p < 0.001$ versus mTBI group.

and butyrate in feces (Fig 6 A). Pearson correlation analysis between SCFAs and differentially abundant gut microbiota revealed significant correlations between butyrate, acetate, and specific microbial taxa (Fig 6 A). Subsequent analysis of serum samples confirmed an increase in butyrate levels following sensory stimuli (Fig 6 A). Based on these observations,

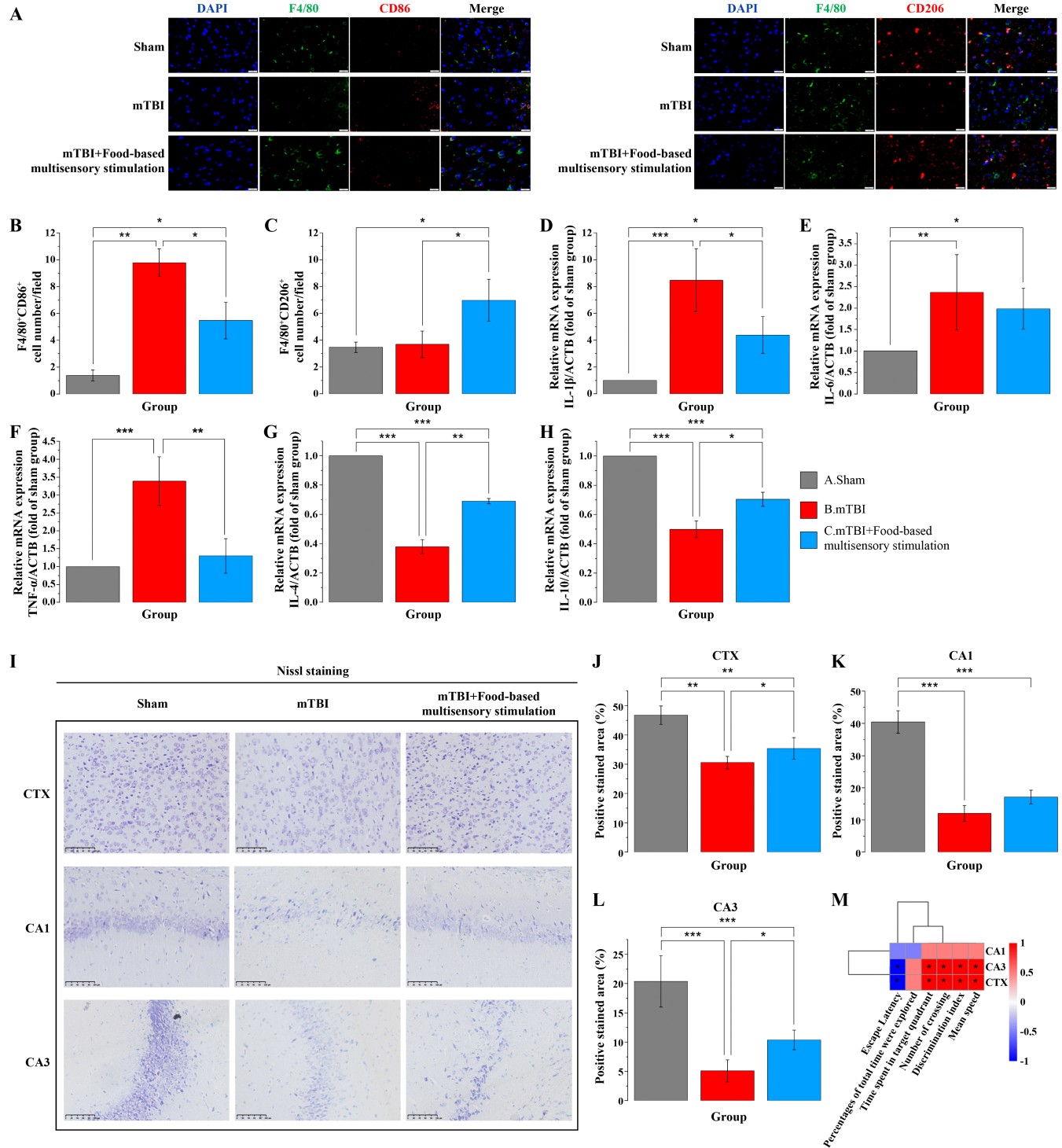

**Fig 5. Food-based multisensory stimulation attenuates neuroinflammation and neuronal loss after mTBI in rats. A.** Representative immunofluorescence images of M1 (F4/80+ CD86+) and M2 (F4/80+ CD206+) macrophages in peri-lesional cerebral cortex (CTX). **B and C.** The quantification of M1 (F4/80+ CD86+) and M2 (F4/80+ CD206+) macrophages in peri-lesional CTX was performed using a minimum of three rats per experimental group, analyzing at least six non-overlapping fields of view per slide (one-way ANOVA with Tukey post hoc test). **D-H.** Quantitative PCR detection of TNF-α, IL-6, IL-1β, IL-4, and IL-10 in the peri-lesional CTX in each group (one-way ANOVA with Tukey post hoc test). **I.** Representative images of Nissl staining

in the peri-lesional CTX, as well as in the hippocampal CA1 and CA3 regions, are presented. **J-L.** Quantitative analysis of Nissl-stained neurons was conducted in the peri-lesional CTX, as well as in the CA1 and CA3 regions of the hippocampus (one-way ANOVA with Tukey post hoc test). **M.** Correlation analysis was conducted between cognitive-related behavioral outcomes in rats and the density of Nissl-stained positive neurons in the peri-lesional CTX, CA1, and CA3 regions (Pearson correlation analysis). N = 8/group, data are presented as mean±SD. *$p < 0.05$, **$p < 0.01$ or ***$p < 0.001$ versus mTBI group.

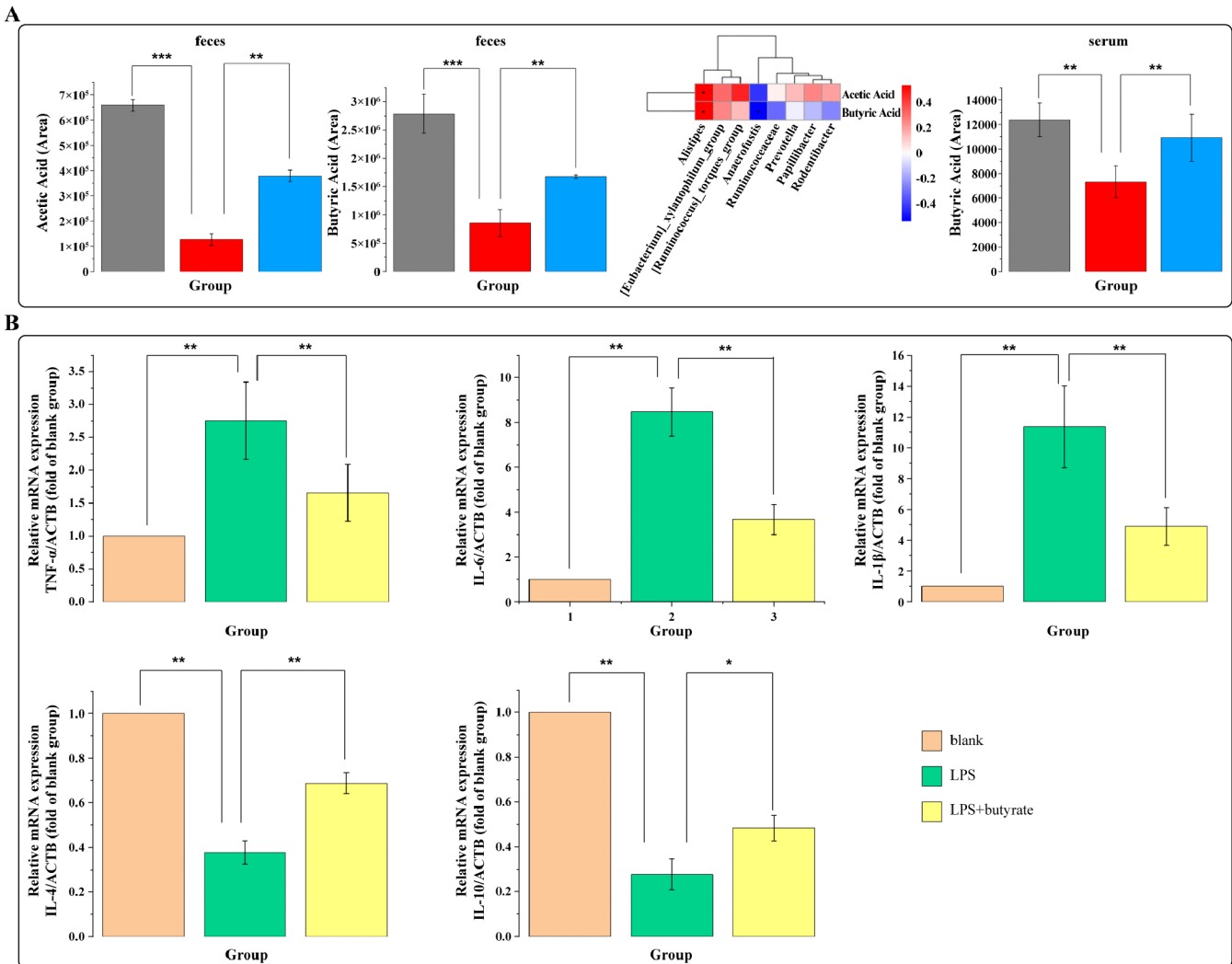

**Fig 6. Butyrate and short-chain fatty acids are associated with reduced neuroinflammation after mTBI. A.** Differences in fecal and serum short-chain fatty acid (SCFA) levels among groups (one-way ANOVA with Tukey's post hoc test), and correlations between the relative abundance of differential intestinal genera and fecal SCFA levels (Pearson correlation analysis). **B.** Quantitative PCR detection of TNF-α, IL-6, IL-1β, IL-4, and IL-10 in the microglia (one-way ANOVA with Tukey post hoc test). N = 8/group, data are presented as mean±SD. *$p < 0.05$, **$p < 0.01$ or ***$p < 0.001$ versus LPS group.

we hypothesized that sensory stimuli may modulate the gut microbiota in mTBI rats and be associated with enhanced butyrate production. Given that butyrate can cross both the intestinal and blood-brain barriers, and has been reported to exert anti-inflammatory effects, it may contribute to that improvement of cognitive impairment in mTBI rats. To validate this hypothesis, in vitro experiments using microglial cells demonstrated that butyrate significantly reduced the mRNA levels

of pro-inflammatory cytokines TNF-α, IL-6, and IL-1β, while increasing the mRNA levels of the anti-inflammatory cytokines IL-4 and IL-10 (Fig 6 B).

## Discussion

CI following mTBI is a significant clinical issue affecting patients' quality of life. In recent years, the role of the gut microbiota in regulating brain inflammation and function through the "gut–brain axis" has gradually been revealed [29]. This study demonstrates that a food-based multisensory stimulation intervention improves cognitive deficits in rats with mTBI. Moreover, our findings suggest that this intervention is associated with changes in the gut microbiota and its metabolites— short-chain fatty acids, particularly butyrate—which may be linked to reduce intestinal and neuroinflammation.

First, we assessed the effects of multisensory stimulation on the cognitive functions of rats with mTBI using the Morris water maze and novel object recognition tests. The results showed that mTBI rats had significant deficits in spatial learning and memory, as evidenced by increased escape latency, fewer platform crossings, and reduced time spent in the target quadrant. However, these cognitive deficits were significantly ameliorated following a food-based multisensory stimulation intervention. In addition, the discrimination index increased to control-like levels in the novel object recognition test, indicating that food-based multisensory stimulation mitigated recognition memory deficits in mTBI rats. The cognitive benefits of food-based multisensory stimulation have been reported in several studies. For example, one study found that overweight and obese participants who completed a food-selection task involving visual exposure to food images before testing performed significantly better on a cognitive test than those who completed the cognitive test without prior exposure [30]. Another study reported that seeing and tasting foods presented in a variety of colors positively influences cognition in patients with mild cognitive impairment or mild dementia [13]. These effects may be attributable to increased orexigenic peptide secretion induced by food-based sensory stimulation; signaling by insulin, orexin, and leptin promotes hippocampal synaptic plasticity and enhances cognitive performance [31]. In addition, a study examining the effects of multisensory stimulation on adult male offspring of thyroidectomized dams showed that multisensory stimulation restored short-term memory [32]. In older populations, multisensory training programs significantly improve cognitive and functional performance [33]. The mechanisms underlying the cognitive benefits of multisensory stimulation primarily involve enhancing neuroplasticity, promoting synaptic remodeling, and modulating neurotransmitter levels via multiple pathways [34,35]. Mild traumatic brain injury often leads to dysregulation of the gut microbiota, which perturbs the gut–brain axis and triggers neuroinflammation, leading to cognitive deficits. However, few studies have investigated the potential role of the gut microbiota and its metabolites in ameliorating or preventing cognitive deficits after mTBI.

To further investigate how multisensory stimulation modulates CI after mTBI via the gut microbiota and their metabolites, we conducted 16S rRNA sequencing analysis on fecal samples from mTBI rats. The results revealed that mTBI induced structural changes in the gut microbiota, characterized by a decrease in the relative abundance of *Ruminococcaceae* and an increase in the relative abundance of *Alistipes* and *Prevotella*. *Ruminococcaceae* is considered a potentially beneficial gut bacterium, known for promoting fiber degradation and SCFA production, enhancing intestinal barrier function, and modulating immunity [36]. Other classical butyrate-producing genera, such as Roseburia and Faecalibacterium, have also been reported to exert anti-inflammatory effects in the gut and central nervous system [37–39]. Although the taxonomic resolution of our 16S rRNA sequencing did not allow us to fully resolve all butyrate-producing genera, the reduction in Ruminococcaceae observed in mTBI rats is consistent with the reduced butyric acid levels detected in this group, whereas the partial restoration of Ruminococcaceae populations in the multisensory stimulation group parallels the elevated butyric acid levels and the reduced intestinal and cerebral inflammation observed in these animals. In contrast, *Alistipes* and *Prevotella* are potential conditional pathogens that can cause infections or even sepsis under conditions of immunosuppression or gut dysbiosis [40,41]. However, multisensory stimulation intervention partially reversed this dysbiosis, increasing the relative abundance of beneficial bacteria and reducing the abundance of potentially harmful bacteria. These alterations in microbiota structure may be related to the modulation of neuroendocrine–immune networks by

multisensory stimulation. The vagus nerve transmits signals from the upper gastrointestinal tract and other visceral organs to the brainstem, while its efferent fibers regulate intestinal peristalsis, secretion, and local immune status, thereby influencing microbiota homeostasis [42,43]. Multiple sensory stimuli can significantly alter HPA axis activity, including changes in adrenocorticotropic hormone (ACTH) and corticosterone levels under baseline and stress conditions in animal models [44], and the resulting changes in stress hormones affect gut barrier function, microbiota composition, and peripheral immune responses [45,34]. In concert with the actions of SCFAs, these neuroendocrine adaptations support a systemic anti-inflammatory milieu that promotes M2 microglial polarization and neuronal protection in vulnerable brain regions after mTBI.

In addition to these gut–brain axis–mediated mechanisms, our findings suggest the possible involvement of the gut–liver–brain axis. After absorption in the colon, short-chain fatty acids first enter the liver via the portal circulation. In the liver, within the context of the gut-liver axis, short-chain fatty acids can regulate energy metabolism and lipid processing in hepatocytes and interact with related pathways, such as bile acid synthesis and conversion, thereby affecting systemic lipid and glucose homeostasis, as well as the production of circulating inflammatory mediators [46,47]. Thus, reduced intestinal permeability and a more favorable short-chain fatty acid profile may lower hepatic endotoxin exposure, attenuate hepatic inflammation and downstream pro-inflammatory signaling, and thereby indirectly mitigate persistent neuroinflammation after traumatic brain injury. Although we did not directly assess liver structure or function, the elevated serum butyrate levels and reduced pro-inflammatory cytokine expression in the gut and brain are consistent with a broader systemic anti-inflammatory effect mediated via the gut–liver–brain axis.

The metabolites of gut microbiota, particularly SCFAs such as butyrate, have been shown to possess anti-inflammatory and neuroprotective properties [48]. In this study, we found that multisensory stimulation significantly increased butyrate levels in both feces and serum. Butyrate can cross the intestinal barrier and the blood-brain barrier, exerting anti-inflammatory effects, reducing intestinal and brain inflammation, and improving intestinal barrier structure. Through qPCR analysis, we further confirmed that multisensory stimulation reduced the mRNA levels of pro-inflammatory cytokines such as TNF-α and IL-1β in the gut and brain, while increasing the mRNA levels of anti-inflammatory cytokines such as IL-4 and IL-10 in the brain. The microglial inflammation model also demonstrated the anti-inflammatory effects of butyrate in the brain. These findings suggest that food-based multisensory stimulation may reduce intestinal and brain inflammation after mTBI by promoting butyrate production; however, this interpretation remains preliminary. Notably, butyrate was measured in both fecal and serum samples as relative levels rather than absolute concentrations (μM); therefore, the 5 mM in vitro exposure cannot be directly compared with physiological butyrate levels and may exceed them. Future studies with validated absolute SCFA quantification and physiologically relevant dosing are warranted.

To further validate the anti-inflammatory effects of butyrate in the brain, we assessed the impact of multisensory stimulation on brain inflammation and neuronal damage using immunofluorescence staining and Nissl staining. The results showed that multisensory stimulation promoted the polarization of microglia from the pro-inflammatory M1 phenotype to the anti-inflammatory M2 phenotype, reducing the number of M1 microglia while increasing the number of M2 microglia. Nissl staining results indicated that multisensory stimulation had a protective effect on neuronal damage following mTBI, particularly in the cortex surrounding the injury and hippocampal CA3 regions. These regions significantly influence cognitive functions such as memory, learning, spatial navigation, and episodic memory [49,50]. Additionally, our results demonstrated that neuronal density in these regions positively correlated with cognitive behavioral indices (e.g., Discrimination index, platform crossings, and time spent in the target quadrant), suggesting that increased neuronal density in these areas is closely associated with improved cognitive function. The differential protective effects observed in various subregions of the hippocampus—that is, neurons in area CA3 and the cortex surrounding the injury were relatively better protected, whereas no significant improvement was seen in area CA1—suggest that the effects of multisensory stimulation and intestinal metabolites may be brain-region specific. Previous studies have shown that CA1 pyramidal neurons are highly sensitive to ischemia, excitotoxicity, and a variety of secondary injury cascades, whereas CA3 and certain cortical

regions have relatively greater potential for structural plasticity and compensatory remodeling [51,52]. In addition, the role of gut-derived SCFAs in the brain may also exhibit regional differences. It has been suggested that SCFA receptors (e.g., GPR41, GPR43), as well as transporter proteins mediating SCFA uptake, are not uniformly distributed in expression across brain regions and cell types [53,54]. If CA3 neurons or neurons, astrocytes, and microglia in the cortex are more sensitive to SCFA receptor signaling, then these regions may be more susceptible to anti-inflammatory and neuroprotective effects in the context of elevated systemic levels of butyric acid induced by multisensory stimulation, whereas CA1 may be relatively "underresponsive." Future studies are warranted to further validate these findings at the tissue and cellular levels.

## Translational implications

In this study, multisensory stimulation was associated with alterations in gut microbiota composition, increased butyric acid levels, and reduced inflammation in the gut and brain, supporting the gut microbial–metabolic axis as a potential therapeutic target to improve cognitive function after mTBI. From a translational perspective, future strategies should move beyond non-specific "one-size-fits-all" probiotic or prebiotic supplementation and instead emphasize strain specificity and inter-individual differences. Studies have shown that different strains within the probiotic genera *Lactobacillus* and *Bifidobacterium* exhibit distinct characteristics in terms of adherence, antibiotic tolerance, gastrointestinal biofluid tolerance, and immunomodulatory activity [55]. They also vary in terms of safety. Moreover, different *Lactobacillus* and *Bifidobacterium* strains have been reported to exert pronounced strain-specific effects on intestinal barrier function, immune and inflammatory responses, and the function of distant organs via the gut–hepatic and gut–brain axes, and some strains even exert divergent biological effects among themselves [56,57]. At the same time, individualized microbiome interventions are increasingly recognized as consistent with the concept of Predictive, Preventive and Personalized Medicine (PPPM) [58]. For patients with mTBI and cognitive impairment, future translational research could combine systematic assessment of baseline gut microbiome composition, inflammatory status, and metabolic phenotype with targeted selection or combination of probiotics and prebiotics tailored to these characteristics, with the aim of enhancing neuroprotective, anti-inflammatory metabolic pathways.

On this basis, objective, axis-specific assessments of the gut–brain and gut–liver–brain axes will also be important in future translational research. Clinically, on the one hand, short-chain fatty acid profiles (especially butyric acid, propionic acid, and acetic acid) in feces and blood, as well as systemic inflammation and barrier function indices (e.g., IL-6, TNF-α, C-reactive protein, LPS, and zonulin) that parallel the findings of the present study, could be used to characterize changes in the activity and permeability of the gut microbial–metabolic axis. In parallel, cerebrospinal fluid markers of inflammation and neurological injury (e.g., YKL-40, neurofilament light chain [NfL], soluble TREM2 [sTREM2]), autonomic function indicators (e.g., heart rate variability), and visceral nociceptive assessments incorporating myofascial- and posture-related phenotypes could be integrated into a multimodal evaluation framework for patients with mTBI to quantify gut–brain and gut–liver–brain axis dysfunction and its relationship to central sensitization [59–61]. These multimodal markers may be combined with individualized microbiome interventions for patient stratification, efficacy monitoring, and regimen optimization in future clinical studies.

## Limitations

The present study nevertheless has certain limitations. First, our 7-day intervention focused on the acute to early subacute phase after mTBI, whereas remodeling of the gut microbiota may take several weeks to be fully established. Therefore, we cannot exclude the possibility that a longer food-based multisensory stimulation intervention might result in additional or more enduring cognitive and anti-inflammatory effects. Second, the present study only explored the effects of food-based multisensory stimulation on cognition and gut microbiota in male rats with mTBI. Male animals were chosen primarily to reduce the potential confounding effects of physiological fluctuations associated with the estrous cycle on behavioral

and microbiota outcomes. However, this design may also limit the generalizability of the findings to female rats. Third, the results of the present study suggest a correlation between multisensory stimulation and altered gut microbiota composition, elevated butyrate levels, and improved neurological and inflammatory outcomes, but do not establish a direct causal relationship between butyrate and the observed behavioral improvements.

Future studies are warranted to clarify this causal relationship through several approaches: (1) applying broad-spectrum antibiotics or approaches such as fecal microbiota transplantation (FMT) to deplete or remodel the gut microbiota to test whether the behavioral benefits of multisensory stimulation depend on gut microbes; (2) specifically intervening in butyric acid-associated signaling pathways through pharmacological or genetic means (e.g., short-chain fatty acid receptor antagonists or histone deacetylase [HDAC] activity modulation) to define the mechanistic contribution of butyrate to the effects of multisensory stimulation; and (3) conducting longitudinal time-series studies to determine whether changes in gut microbiota and short-chain fatty acid levels precede improvements in neuroinflammation and cognitive function, rather than merely accompanying them. Such studies will help to elucidate the extent of the causal role of the gut microbial–butyric acid pathway in food-based multisensory stimulation interventions for mTBI.

## Conclusion

This study suggests a potential association pattern whereby multisensory stimulation is linked to reduced intestinal and brain inflammation and improved cognitive impairment after mTBI, in parallel with alterations in gut microbiota and their metabolites, particularly butyrate. Specifically, multisensory stimulation appeared to modulate gut microbial composition and increase butyrate production, which may cross the intestinal and blood–brain barriers to exert anti-inflammatory effects, thereby reducing neuroinflammation and potentially improving cognitive function. However, this mechanistic pathway requires further validation. These findings provide new therapeutic insights for cognitive rehabilitation after mTBI, suggesting that multisensory stimulation may serve as a non-invasive neuromodulation approach with broad clinical application prospects. However, the interaction mechanisms between gut microbiota and the brain are complex, and further in-depth research is needed to fully elucidate the molecular mechanisms and explore the long-term effects of multisensory stimulation on cognitive function.

## Supporting information

**S1 File. Original Uncut Images.**
(PDF)

**S2 File. ARRIVE guideline.**
(PDF)

**S3 File. Raw data.**
(XLSX)

## Acknowledgments

We would like to acknowledge the collaborators and research teams of the study.

## Author contributions

**Conceptualization:** Yuhan Wei.

**Supervision:** Ka Li.

**Writing – original draft:** Yuhan Wei, Qiujing Du.

**Writing – review & editing:** Hanif Ullah, Guangneng Liao, Xue Xiao, Jiang Yao, Qiujing Du, Qijie Li, Ka Li.

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
