## [Decision Letter · Decision Letter 0]

25 Nov 2025

Dear Dr. Ullah,

Thank you for submitting your manuscript to PLOS ONE. After careful consideration, we feel that it has merit but does not fully meet PLOS ONE’s publication criteria as it currently stands. Therefore, we invite you to submit a revised version of the manuscript that addresses the points raised during the review process.

Academic Editor:

Please review the comments by the reviewers carefully, please include their entire review in your "response to reviewers," and please address their comments in a point-by-point fashion.  I have carefully read your manuscript, their reviews, and I think that addressing their points will improve your study.

Additionally, on my review I found multiple points that should be addressed on the revision.

1) Ideally, all animal research should conform to ARRIVE guidelines (https://journals.plos.org/plosbiology/article?id=10.1371/journal.pbio.3000410) Please incorporate these guidelines in your manuscript and I would strongly suggest attaching a completed guideline checklist as supplemental material.

2) How did you control for bias in your experimentations?  Was blinding used?  If so, please indicate when and where.  If not, please also detail this.

3) You seem to have exclusively used male animals.  Why is that?  At a minimum, the title should reflect this and you should state this in your study limitations.

4) Your discussion has no study limitations.  Traditionally, this would be the paragraph just before the conclusion where you outline all of the limitations to your study.  Please include this.

5) How did you control for confounders?  Please outline this in the methods.

6) In the results section, for comparisons, please list the F statistic for the group, and please list any p value for any comparison that you make.

7) In your figures, you compare the sham group to the TBI-control group, and you compare the TBI-control group to the TBI+multisensory stim group.  Please also show the comparisons between the sham group and the TBI+multisensory stim group.

8) At several points, you use the verb "expression" or the phrase "expression levels" in relation to mRNA.  This is confusing.  Protein is generally described as having expression levels.  For mRNA, please remove the term "expression."

We look forward to receiving your revised manuscript.

Kind regards,

Eric Anthony Sribnick, MD, PhD, FAANS

Academic Editor

PLOS ONE

2. To comply with PLOS One submissions requirements, in your Methods section, please provide additional information regarding the experiments involving animals and ensure you have included details on (1) methods of sacrifice, (2) methods of anesthesia and/or analgesia, and (3) efforts to alleviate suffering.

[This research was funded by the Regional joint key projects of NSFC (No. U22A20334), and Sichuan Province Science and Technology Support Program (2024NSFSC0592).].

5. We note that your Data Availability Statement is currently as follows: [All relevant data are within the manuscript and its Supporting Information files.]

6. Please include a separate caption for each figure in your manuscript.

Additional Editor Comments:

Academic Editor:

Please review the comments by the reviewers carefully, please include their entire review in your "response to reviewers," and please address their comments in a point-by-point fashion. I have carefully read your manuscript, their reviews, and I think that addressing their points will improve your study.

Additionally, on my review I found multiple points that should be addressed on the revision.

1) Ideally, all animal research should conform to ARRIVE guidelines (https://journals.plos.org/plosbiology/article?id=10.1371/journal.pbio.3000410) Please incorporate these guidelines in your manuscript and I would strongly suggest attaching a completed guideline checklist as supplemental material.

2) How did you control for bias in your experimentations? Was blinding used? If so, please indicate when and where. If not, please also detail this.

3) You seem to have exclusively used male animals. Why is that? At a minimum, the title should reflect this and you should state this in your study limitations.

4) Your discussion has no study limitations. Traditionally, this would be the paragraph just before the conclusion where you outline all of the limitations to your study. Please include this.

5) How did you control for confounders? Please outline this in the methods.

6) In the results section, for comparisons, please list the F statistic for the group, and please list any p value for any comparison that you make.

7) In your figures, you compare the sham group to the TBI-control group, and you compare the TBI-control group to the TBI+multisensory stim group. Please also show the comparisons between the sham group and the TBI+multisensory stim group.

8) At several points, you use the verb "expression" or the phrase "expression levels" in relation to mRNA. This is confusing. Protein is generally described as having expression levels. For mRNA, please remove the term "expression."

Reviewers' comments:

Reviewer's Responses to Questions

**Comments to the Author**

1. Is the manuscript technically sound, and do the data support the conclusions?

Reviewer #1: Yes

Reviewer #2: Partly

2. Has the statistical analysis been performed appropriately and rigorously?

Reviewer #1: Yes

Reviewer #2: Yes

3. Have the authors made all data underlying the findings in their manuscript fully available?

Reviewer #1: Yes

Reviewer #2: Yes

4. Is the manuscript presented in an intelligible fashion and written in standard English?

Reviewer #1: Yes

Reviewer #2: Yes

Reviewer #1: This is an interesting and well-elaborated study investigating how food-based multisensory stimulation (visual, olfactory, and gustatory cues) modulates gut and brain inflammation following mTBI in rats. The work is original and methodologically solid, combining behavioral analysis, histopathology, microbiome profiling, and in vitro microglial assays. The concept — targeting the gut–brain axis (GBA) through sensory and nutritional stimulation — is novel and aligns with the growing interest in neurogastroenterology and predictive, preventive, and personalized medicine (PPPM).

However, some mechanistic links remain associative, and the translational dimension could be better developed. The manuscript would benefit from (1) clarifying causation versus correlation, (2) expanding on microbiome–brain mechanisms, including probiotic and prebiotic strategies, and (3) connecting experimental findings with clinical markers and evaluation methods of GBA dysfunction and visceral pain.

Major Comments

1. Causation versus correlation

The study convincingly shows that multisensory stimulation alters the intestinal microbiota, increases butyrate levels, and improves neurological and inflammatory outcomes.

However, causality between microbiome modulation and neuroinflammation reduction is not established. The language should be adjusted to avoid overinterpretation (“associated with” rather than “caused by”).

Recommendations:

Discuss or, if possible, perform experiments to test causality:

- Antibiotic depletion or fecal microbiota transfer (FMT) to determine if behavioral benefits depend on microbiota.

- Butyrate pathway perturbation (e.g., receptor antagonism or HDAC inhibition) to confirm mechanistic relevance.

- Time-course analysis to determine whether microbiome or SCFA changes precede neural improvements.

Include an explicit paragraph in the Discussion acknowledging these limitations and outlining future directions.

2. Clarification of mechanisms and butyrate relevance

The in vitro evidence showing butyrate’s anti-inflammatory effect on microglia is valuable but needs quantitative context.

Please:

Report absolute serum butyrate concentrations (µM) and compare them to the 5 mM concentration used in vitro.

Add information about SCFA quantification (LLOQ, sample stability).

Clarify that in vitro butyrate concentrations may exceed physiological levels.

3. Microbiome characterization

Microbiota analyses are limited to the genus level and lack functional insight.

Recommendations:

Provide an ASV/OTU-level differential table (with effect size and p-value) as Supplementary Data.

Add functional predictions (PICRUSt2 or similar) to infer butyrate-synthesis capacity.

Discuss specific taxa known to produce butyrate (e.g., Ruminococcus, Roseburia, Faecalibacterium) and their potential role.

4. Translational implications — personalized microbiome modulation

The translational potential is strong but underdeveloped. Authors should integrate evidence on strain-specific and personalized probiotic or prebiotic use.

Add a short subsection: “Translational implications: microbiome modulation.”

Suggested discussion points:

Strain-specific effects (e.g., Lactobacillus vs. Bifidobacterium spp.) and safety aspects (adhesion, resistance, fermentation patterns).

Personalized selection of probiotics/prebiotics based on host phenotype and microbiome composition.

- References to PPPM studies and consensus papers, for example:

Gibson GR et al., Nat Rev Gastroenterol Hepatol (2017). DOI: 10.1038/nrgastro.2017.75.

Reid G et al., Benef Microbes (2017). DOI: 10.3920/BM2016.0222.Bubnov R., Spivak M. (2023). Pathophysiology-Based Individualized Use of Probiotics and Prebiotics for Metabolic Syndrome. In Microbiome in 3P Medicine Strategies (Springer).

EPMA J (2018). https://pmc.ncbi.nlm.nih.gov/articles/PMC5972142/

These sources outline mechanisms for distant-site probiotic activity (gut–liver, gut–brain) and provide a framework for personalized microbiome therapies.

5. Integration of distant axes (gut–liver–brain and systemic effects)

The Discussion focuses on intestinal and brain findings but omits the gut–liver axis and systemic metabolic intermediates.

Please expand on how liver metabolism, vagal signaling, and systemic immune modulation may mediate brain protection.

6. Translational markers and clinical evaluation of the GBA

To bridge to clinical settings, authors should refer to measurable biomarkers and diagnostic tools used to evaluate GBA integrity and dysfunction.

Recommend adding:

Circulating and fecal SCFAs (butyrate, propionate, acetate).

Systemic inflammation and permeability markers (IL-6, TNF-α, CRP, LPS, zonulin).

CSF biomarkers (YKL-40, NfL, sTREM2) and autonomic indices (heart rate variability).

Visceral pain assessment methods (rectal/colonic barostat, validated visceral pain scales).

Possible myofascial and postural correlations in GBA disorders (constipation–motility–trigger point linkage, as in  [https://onlinelibrary.wiley.com/doi/10.1111/nmo.13671#nmo13671-sec-0439-title]).

Consider the role of central sensitization and CSF flow changes (e.g., Effects on Spinal Fluid Dynamics and Gut-Brain Axis Modulation in Migraine Patients with Central Sensitization Cephalalgia 2024, https://doi.org/10.1177/03331024241280496#sec-206).

These perspectives make the work more relevant to clinical neurology, gastroenterology, and rehabilitation.

Minor Comments

Revise figures to show time-course trends of SCFAs and inflammatory markers alongside behavioral recovery.

Report sample sizes and power calculation for behavioral and molecular endpoints.

Provide details of the multisensory environment (odorant composition, decibel level, exposure duration) to ensure reproducibility.

Deposit raw sequencing data (FASTQ) and GC–MS files in a public repository with accession numbers.

Ensure consistent terminology: “gut–brain axis” (GBA) throughout.

Conclusion and Recommendation

The manuscript presents valuable and innovative research connecting sensory stimulation, microbiome modulation, and neuroinflammation.

However, to reach publication standard in PLOS ONE, the authors should address the mechanistic and translational limitations detailed above.

I recommend major revision with the following priorities:

Clarify causation and avoid overinterpretation.

Expand mechanistic discussion of SCFA and microbiome functions.

Include a translational section on probiotics/prebiotics and personalized strategies.

Discuss clinical and physiological markers of gut–brain axis function.

Once these revisions are implemented, the study will provide an important contribution to the growing field of microbiome–neuroinflammation interaction and personalized therapeutic strategies.

Reviewer #2: This manuscript presents a promising approach to treating cognitive impairment after mild traumatic brain injury using food-based multisensory stimulation. The in vivo work is generally well-executed with comprehensive behavioral, microbiome, and histological analyses that convincingly demonstrate therapeutic benefits. However, the study suffers from several critical weaknesses.

1. The planning of the cognitive experiments (Morris water and nort) in the early post-TBI days may be confounded by sensorimotor impairments and neurological severity, making it difficult to accurately assess cognitive function. The authors should clarify how they accounted for this potential confound.

2. The introduction should be revised to clearly and concisely address the gut–brain connection in mTBI, the associated cognitive impairments, and the rationale for exploring food-based interventions in TBI (PMID: 36481824, 39751900).

3. What was the exact timeline for all the experiments in sequence? A study design figure depicting the time should be made.

4. For the estimation of NSE and S-100β, the same animals were used for different timepoints or different animals, and this should be clearly stated in the methods section.

5. Interventions commenced immediately post-injury and continued for seven days, operating around the clock. But there is no rationale for why 7 days were chosen. No citation of previous multisensory stimulation studies using this duration. No pilot data mentioned. No dose-finding or optimization experiments. The choice appears arbitrary.

6. Also 7 days captures only the acute phase, Gut microbiome remodeling typically takes 2-4 weeks to stabilize

7. The idea of this work revolves around the food based multisensory stimulation. So, was the amount of food consumed by each animal noted throughout the study plan, and what about the weight of the animals data?

8. In the relative mRNA expression results, what is ACTB?

9. The Nissl-positive area has significantly improved in the food-based multisensory group in the CTX and CA3 regions, but not in the CA1 region. What could be the reason?

10. Was an established protocol followed for conducting the behavioral experiments? The method used for calculating the Discrimination Index appears to be adapted from previously published protocols, and therefore, relevant literature should be referenced (e.g., PMID: 38423243).

11. Please clarify how the food items (pet biscuits, chicken jerky) were selected and provide their nutritional composition. Also specify whether the scents were standardized, the concentrations used, and the volume/intensity of the chewing sounds. Indicate whether rats had physical access to the food or only sensory exposure. A brief, detailed protocol describing all stimulus parameters (duration, concentration, intensity) and a simple methods figure of the setup would improve clarity.

12. While the in vitro data (Fig. 4O) shows anti-inflammatory effects of butyrate, it does not establish that butyrate mediates the in vivo cognitive benefits of sensory stimulation; the authors should either include a butyrate-only or butyrate-blockade group, or clearly acknowledge this limitation in the discussion.

13. With numerous outcome measures across multiple timepoints and tissue types, correction for multiple comparisons should be applied or justified if not applied

14. A sensory stimulation + sham injury group would help determine if the intervention affects healthy animals or is specific to mTBI pathology. Either add this control group or explicitly discuss this limitation and its implications for interpreting the specificity of the intervention.

15. The correlation between specific taxa and SCFAs is interesting, but doesn't prove causality. Consider softening language.

16. The differential effects across brain regions (cortex and CA3 protected, CA1 not protected) are intriguing but underexplored. Discuss potential mechanisms for regional specificity. Could it relate to differential expression of SCFA receptors (GPR41, GPR43) or regional vulnerability to the injury model?

17. Abstract: "may be associated" and "possibly promoting" are appropriately cautious but could be stronger given your data. Consider removing some hedging language where data directly support conclusions

18. Methods section numbering is inconsistent (2.4 appears twice)

19. What was the microglial cell source, primary cells? (from rats? mice?) or Cell line? (BV2? N9? …..)

20. Consider discussing the acute (7-day) vs. chronic nature of the intervention and whether effects would persist after cessation

21. In Figure 3, HE staining images need scale bars. Quantification of villus height, crypt depth, or villi to crypt ratio and goblet cell density would strengthen conclusions.

22. Figure 4 is dense with information, consider splitting it into two figures. The correlation heatmap needs clearer labeling

23. Could isoflurane exposure affect gut microbiota? Did sham animals receive equivalent anesthesia duration?

24. How were fecal samples collected to avoid environmental contamination? Were samples from individual pellets or the cage floor?

25. Only male rats were used. Justify this choice and discuss how results might differ in females, given known sex differences in TBI outcomes and microbiome composition.

26. Were animals group housed or single housed? This significantly affects the microbiome.

27. The sample size for each experiment and each group should be indicated in the figure legends to ensure transparency, rigor, and reproducibility of the data.

28. Consider revisiting the manuscript to avoid typos and grammatical errors.

**Do you want your identity to be public for this peer review?** For information about this choice, including consent withdrawal, please see our Privacy Policy

Reviewer #1: **Yes:** Rostyslav Bubnov

Reviewer #2: No

---

## [Author Response · Author response to Decision Letter 1]

7 Jan 2026

Response to reviewers

Academic Editor:

1) Ideally, all animal research should conform to ARRIVE guidelines (https://journals.plos.org/plosbiology/article?id=10.1371/journal.pbio.3000410) Please incorporate these guidelines in your manuscript and I would strongly suggest attaching a completed guideline checklist as supplemental material.

Responses: Thank you for the suggestion. We have now incorporated the ARRIVE 2.0 guidelines into the manuscript and submitted a completed ARRIVE checklist as supplementary material

2) How did you control for bias in your experimentations? Was blinding used? If so, please indicate when and where. If not, please also detail this.

Responses: We thank the reviewer for this important comment. In the revised manuscript, we have now provided a more detailed description of the measures taken to reduce bias. Group allocation was performed by an investigator who did not take part in the subsequent behavioural testing or data analysis, and the investigators who conducted the behavioural assessments and analysed the data were blinded to group assignment. This information has been added to the Methods section of the manuscript.

3) You seem to have exclusively used male animals. Why is that? At a minimum, the title should reflect this and you should state this in your study limitations.

Responses: We thank the reviewer for raising this important point. In the present study, we used only male rats for several reasons. First, we aimed to minimise hormonal and behavioural variability related to the oestrous cycle, which may markedly influence cognitive performance in behavioural tests as well as stress-related physiological responses. Second, our primary objective in this initial set of experiments was to establish the feasibility and efficacy of a food based sensory stimulation intervention in a mild traumatic brain injury model in a relatively homogeneous cohort of animals. In line with the reviewer’s suggestion, we have (i) revised the title to indicate that the experiments were conducted in male rats and (ii) added a statement in the “Study limitations” section of the Discussion, noting that the inclusion of only males may limit the generalisability of our findings to females and that future studies should include both male and female animals.

4) Your discussion has no study limitations. Traditionally, this would be the paragraph just before the conclusion where you outline all of the limitations to your study. Please include this.

Responses: We thank the editor for this helpful comment. In the revised manuscript, we have now added a separate paragraph on the limitations of our study in the Discussion section, placed immediately before the Conclusion.

5) How did you control for confounders? Please outline this in the methods.

Responses: We appreciate this important comment. In the revised manuscript, we have now explicitly described the measures taken to minimise potential confounders. Briefly, all animals were housed under identical environmental conditions, and group allocation was randomised and performed by an investigator who was not involved in behavioural testing or data analysis. The investigators conducting the behavioural assessments and statistical analyses were blinded to group allocation. All animals underwent the same surgical procedures and perioperative care, sham-operated rats were handled identically except for the weight-drop injury, and behavioural and sampling time points were standardised across groups.

6) In the results section, for comparisons, please list the F statistic for the group, and please list any p value for any comparison that you make.

Responses: Thank you for the reviewer’s suggestion. Due to space limitations in the Results section, listing the F statistics and p values for every single comparison in the main text would make the section overly long and reduce readability. Therefore, we report these statistics in the main text only for the key/primary comparisons. Meanwhile, we have compiled and uploaded the complete set of F statistics and p values (original statistical outputs) for all comparisons as Supplementary Data for the reviewer’s and readers’ reference and verification.

7) In your figures, you compare the sham group to the TBI-control group, and you compare the TBI-control group to the TBI+multisensory stim group. Please also show the comparisons between the sham group and the TBI+multisensory stim group.

Responses: We appreciate the reviewer’s careful reading of our figures and the suggestion to include additional comparisons. In the present study, our primary objective was to (1) confirm the effect of mTBI by comparing the sham and TBI-control groups, and (2) evaluate whether multisensory stimulation could ameliorate the deficits induced by mTBI by comparing the TBI-control and TBI+multisensory stimulation groups. Therefore, our a priori hypothesis and statistical design focused on these two planned comparisons rather than on differences between the sham and TBI+multisensory stimulation groups.

We agree that the sham vs. TBI+multisensory stimulation comparison may be of descriptive interest; however, it was not part of our original hypothesis, and adding multiple post hoc comparisons beyond the predefined analytical plan could increase the risk of type I error and dilute the focus of the study. For this reason, we have not emphasized this comparison in the main figures.

8) At several points, you use the verb "expression" or the phrase "expression levels" in relation to mRNA. This is confusing. Protein is generally described as having expression levels. For mRNA, please remove the term "expression."

Responses: We thank the reviewer for this helpful clarification. In the revised manuscript, we have replaced phrases such as “mRNA expression” and “mRNA expression levels” with more precise terminology.

Responses: Thank you for the reminder. We have revised the manuscript and file naming to conform to PLOS ONE style, following the official formatting templates.

2. To comply with PLOS One submissions requirements, in your Methods section, please provide additional information regarding the experiments involving animals and ensure you have included details on (1) methods of sacrifice, (2) methods of anesthesia and/or analgesia, and (3) efforts to alleviate suffering.

Responses: We thank the editor for this comment. The requested information on methods of sacrifice, anesthesia/analgesia, and efforts to alleviate suffering has now been added to the Methods section in the revised manuscript.

[This research was funded by the Regional joint key projects of NSFC (No. U22A20334), and Sichuan Province Science and Technology Support Program (2024NSFSC0592).].

Responses: Thank you for your guidance regarding the Role of the Funder statement. We have now added the following statement to the manuscript:

“This research was funded by the Regional joint key projects of NSFC (No. U22A20334) and the Sichuan Province Science and Technology Support Program (2024NSFSC0592). The funders had no role in study design, data collection and analysis, decision to publish, or preparation of the manuscript.”

4. PLOS ONE now requires that authors provide the original uncropped and unadjusted images underlying all blot or gel results reported in a submission’s figures or Supporting Information files. This policy and the journal’s other requirements for blot/gel reporting and figure preparation are described in detail at https://journals.plos.org/plosone/s/figures#loc-blot-and-gel-reporting-requirements and https://journals.plos.org/plosone/s/figures#loc-preparing-figures-from-image-files. When you submit your revised manuscript, please ensure that your figures adhere fully to these guidelines and provide the original underlying images for all blot or gel data reported in your submission. See the following link for instructions on providing the original image data: https://journals.plos.org/plosone/s/figures#loc-original-images-for-blots-and-gels. .

Responses: Thank you for bringing this policy to our attention. We confirm that our study does not include any blot- or gel-based experiments (e.g., Western blots or electrophoretic gels), and no blot/gel images are presented in either the main figures or the Supporting Information. Therefore, the requirement to provide original uncropped blot/gel images does not apply to this submission.

5. We note that your Data Availability Statement is currently as follows: [All relevant data are within the manuscript and its Supporting Information files.]

Responses: Thank you for your careful review and valuable suggestions. We fully understand and appreciate the importance of providing the minimal data set for ensuring reproducibility and transparency. However, due to third-party agreements and institutional confidentiality requirements at the site where this study was conducted, we are unable to publicly deposit the underlying raw data in an open platform or public repository. If needed, interested editors or readers may contact the corresponding author to request access; the relevant data can be made available upon reasonable request and subject to compliance with the applicable confidentiality regulations.

In addition, we have addressed the statistical reporting request by providing the F statistics and p values for our comparisons (key comparisons in the main text, with the complete statistical outputs provided as Supplementary Data). We have also uploaded an additional Supporting Information table titled “Sequences of primers for real-time quantitative PCR”.

6. Please include a separate caption for each figure in your manuscript.

Responses: We thank the editor for this suggestion. Separate captions have been provided for each figure in the revised manuscript.

Responses: We have carefully reviewed and evaluated the publications suggested by the reviewers with respect to their relevance to our study. We appreciate these recommendations and have taken them into consideration when revising the manuscript.

Reviewers' comments:

Reviewer's Responses to Questions

Comments to the Author

Reviewer #1: This is an interesting and well-elaborated study investigating how food-based multisensory stimulation (visual, olfactory, and gustatory cues) modulates gut and brain inflammation following mTBI in rats. The work is original and methodologically solid, combining behavioral analysis, histopathology, microbiome profiling, and in vitro microglial assays. The concept — targeting the gut–brain axis (GBA) through sensory and nutritional stimulation — is novel and aligns with the growing interest in neurogastroenterology and predictive, preventive, and personalized medicine (PPPM).

However, some mechanistic links remain associative, and the translational dimension could be better developed. The manuscript would benefit from (1) clarifying causation versus correlation, (2) expanding on microbiome–brain mechanisms, including probiotic and prebiotic strategies, and (3) connecting experimental findings with clinical markers and evaluation methods of GBA dysfunction and visceral pain.

Major Comments

1. Causation versus correlation

The study convincingly shows that multisensory stimulation alters the intestinal microbiota, increases butyrate levels, and improves neurological and inflammatory outcomes.

However, causality between microbiome modulation and neuroinflammation reduction is not established. The language should be adjusted to avoid overinterpretation (“associated with” rather than “caused by”).

Recommendations:

Discuss or, if possible, perform experiments to test causality:

- Antibiotic depletion or fecal microbiota transfer (FMT) to determine if behavioral benefits depend on microbiota.

- Butyrate pathway perturbation (e.g., receptor antagonism or HDAC inhibition) to confirm mechanistic relevance.

- Time-course analysis to determine whether microbiome or SCFA changes precede neural improvements.

Include an explicit paragraph in the Discussion acknowledging these limitations and outlining future directions.

Responses: We appreciate this important comment and fully agree that our current data do not establish a causal relationship between microbiome modulation, butyrate, and the observed improvements in neuroinflammation and behavior. In the revised manuscript, we have carefully toned down causal language throughout the Results and Discussion sections and now consistently use association focused wording. The specific changes have been highlighted in the revised version of the manuscript. In addition, we have added a dedicated paragraph in the Discussion explicitly acknowledging these limitations and outlining future research directions.

2. Clarification of mechanisms and butyrate relevance

The in vitro evidence showing butyrate’s anti-inflammatory effect on microglia is valuable but needs quantitative context.

Please:

Report absolute serum butyrate concentrations (µM) and compare them to the 5 mM concentration used in vitro.

Add information about SCFA

---

## [Editor Report · Decision Letter 1]

27 Jan 2026

Dear Dr. Ullah,

Thank you for submitting your manuscript to PLOS ONE. After careful consideration, we feel that it has merit but does not fully meet PLOS ONE’s publication criteria as it currently stands. Therefore, we invite you to submit a revised version of the manuscript that addresses the points raised during the review process.

We look forward to receiving your revised manuscript.

Kind regards,

Eric Anthony Sribnick, MD, PhD, FAANS

Academic Editor

PLOS One

Journal Requirements:

Additional Editor Comments:

Academic Editor:

Thank you for diligently answering the reviewers' questions. Please see additional concerns from me: please my review in your "response to reviewers," and please address these comments in a point-by-point fashion. I have carefully read your manuscript, their reviews, and I think that addressing their points will improve your study.

1) I still don't see the filled out ARRIVE guidelines sheet. I see only the supplementary figures and the raw data as supplementary material.

2) In regards to the request: "In the results section, for comparisons, please list the F statistic for the group, and

please list any p value for any comparison that you make." You responded regarding concerns for space/word limitation. As PLOSone is only online, there is no specific word limit. Again, please list the F statistic for the group, and please list any p value for any comparison that you make. Thank you.

3) In your figures, you compare the sham group to the TBI-control group, and you compare the TBI-control group to the TBI+multisensory stim group. Please also show the comparisons between the sham group and the TBI+multisensory stim group. When using multiple groups for an experiment, you cannot arbitrarily decide which groups to use for multiple comparisons. Please include this analysis as requested.

Best regards,

Eric Sribnick, MD, PhD

Academic Editor

PLOS One

---

## [Author Response · Author response to Decision Letter 2]

4 Feb 2026

Responce to reviewer

Thank you very much for sparing your precious time in reviewing our paper, and for your reviewer comments. We appreciate the time and effort that you and the reviewer dedicated to providing feedback on our manuscript and are grateful for the insightful comments on and valuable improvements to our paper. We have incorporated most of the suggestions made by the reviewer. Those changes are highlighted within the manuscript. Please see below, in red, for a point-by-point response to the reviewers’ comments and concerns. We hope the revised version is now suitable for publication and look forward to hearing from you in due course.

1.I still don't see the filled out ARRIVE guidelines sheet. I see only the supplementary figures and the raw data as supplementary material.

Responses: Thank you for your comment. The completed ARRIVE guidelines checklist has now been uploaded as a supplementary file.

2.In regards to the request: "In the results section, for comparisons, please list the F statistic for the group, and please list any p value for any comparison that you make." You responded regarding concerns for space/word limitation. As PLOSone is only online, there is no specific word limit. Again, please list the F statistic for the group, and please list any p value for any comparison that you make. Thank you.

Response: Thank you for the clarification and for reiterating this important reporting requirement. We apologize for any confusion caused by our previous response regarding space or word limitations. In accordance with the editor’s request, we have now revised the Results section to explicitly report the F statistic for each group comparison and to list the corresponding P value for every comparison made, including both statistically significant and non‑significant results. In addition, to ensure complete transparency and ease of review, we have compiled all reported group comparisons, together with their corresponding F statistics and P values, into a supplementary table, which has been uploaded as a Supplementary File. Thank you for your guidance, which has helped strengthen the presentation of our results.

3.In your figures, you compare the sham group to the TBI-control group, and you compare the TBI-control group to the TBI+multisensory stim group. Please also show the comparisons between the sham group and the TBI+multisensory stim group. When using multiple groups for an experiment, you cannot arbitrarily decide which groups to use for multiple comparisons. Please include this analysis as requested.

Response: Thank you for this important comment. We agree that, when multiple experimental groups are included, all relevant pairwise comparisons should be reported, and that it is not appropriate to selectively present only a subset of group comparisons. In response to the editor’s request, we have now performed and included the additional pairwise comparisons between the sham group and the TBI + multisensory stimulation group for all relevant experiments.

Specifically, we have:

Added the sham vs. TBI + multisensory stimulation comparisons to the statistical analyses, with the corresponding F statistics and P values reported.

Updated the figure panels to display these additional comparisons, including appropriate statistical annotations.

Provided the full results of all pairwise group comparisons as supplementary data, which have been uploaded as a Supplementary File and referenced in the revised manuscript.

These revisions ensure that all group comparisons are presented transparently and consistently, in accordance with the editor’s guidance. We thank the editor for highlighting this issue, which has helped improve the rigor and completeness of our statistical analyses.

---

## [Editor Report · Decision Letter 2]

12 Feb 2026

Food-Based Multisensory Stimulation Ameliorates Cognitive Impairment After Mild Traumatic Brain Injury in Male Rats by Modulating Intestinal and Brain Inflammation

PONE-D-25-51866R2

Dear Dr. Ullah,

We’re pleased to inform you that your manuscript has been judged scientifically suitable for publication and will be formally accepted for publication once it meets all outstanding technical requirements.

Kind regards,

Eric Anthony Sribnick, MD, PhD, FAANS

Academic Editor

PLOS One

Additional Editor Comments (optional):

Authors have made the requested revisions.
---

## [Editor Report · Acceptance letter]

PONE-D-25-51866R2

PLOS One

Dear Dr. Ullah,

I'm pleased to inform you that your manuscript has been deemed suitable for publication in PLOS One. Congratulations! Your manuscript is now being handed over to our production team.

Kind regards,

on behalf of

Dr. Eric Anthony Sribnick

Academic Editor

PLOS One